# Dietary Patterns in Runners with Gastrointestinal Disorders

**DOI:** 10.3390/nu13020448

**Published:** 2021-01-29

**Authors:** Kelly Anne Erdman, Kim Wagner Jones, Robyn F. Madden, Nancy Gammack, Jill A. Parnell

**Affiliations:** 1Canadian Sport Institute, Calgary, AB T3B 6B7, Canada; kerdman@csicalgary.ca; 2Helios Wellness Centres, Calgary, AB T2N 4Z6, Canada; kimjonesrd@gmail.com; 3Health and Physical Education, Mount Royal University, Calgary, AB T3E 6K6, Canada; rmadd656@mtroyal.ca (R.F.M.); ngamm501@mtroyal.ca (N.G.); jparnell@mtroyal.ca (J.A.P.)

**Keywords:** runners, inflammatory bowel disease (IBD), irritable bowel syndrome (IBS), heartburn/reflux, dietary restrictions, dietary patterns, exercise-induced gastrointestinal symptoms, runner’s gut

## Abstract

Individuals with inflammatory bowel disease (IBD), irritable bowel syndrome (IBS) and reflux frequently experience gastrointestinal symptoms (GIS), potentially enhanced by high-intensity running. Food avoidances, food choices, and GIS in runners with IBS/IBD (*n* = 53) and reflux (*n* = 37) were evaluated using a reliability and validity tested questionnaire. Comparisons to a control group of runners (*n* = 375) were made using a Fisher’s Exact test. Runners with IBS/IBD experienced the greatest amount of exercise-induced GIS followed by those with reflux. Commonly reported GIS were stomach pain/cramps (77%; 53%), bloating (52%; 50%), intestinal pain/cramps (58%; 33%), and diarrhea (58%; 39%) in IBS/IBD and reflux groups respectively. In the pre-race meal, those with IBS/IBD frequently avoided milk products (53%), legumes (37%), and meat (31%); whereas, runners with reflux avoided milk (38%), meat (36%), and high-fibre foods (33%). When considering food choices pre-race, runners with IBS/IBD chose grains containing gluten (40%), high fermentable oligo-, di-, mono-saccharides and polyols (FODMAP) fruits (38%), and water (38%). Runners with reflux chose water (51%), grains containing gluten (37%), and eggs (31%). In conclusion, while many runners with IBS/IBD and reflux are avoiding trigger foods in their pre-race meals, they are also consuming potentially aggravating foods, suggesting nutrition advice may be warranted.

## 1. Introduction

Running is one of the most strenuous forms of exercise on the gastrointestinal (GI) tract [1], with prevalence of gastrointestinal symptoms (GIS) exceeding 80% in several studies [2,3,4]. Exercise-induced stress occurs due to adaptations in two physiological pathways affecting the integrity and function of the GI tract [1]. Upon initiating exercise, the circulatory-gastrointestinal pathway redistributes blood flow away from the GI tract, and towards muscles and systems of thermoregulation, ultimately leading to splanchnic ischemia in the GI tract. Additionally, in the neuroendocrine-GI pathway, exercise activates the sympathetic nervous system and the release of stress hormones, which alters gut motility and transit time, and results in the decreased ability to absorb nutrients. Together these two pathways lead to localized inflammation in the GI tract, intestinal translocation of bacteria, and ultimately systemic inflammation [1,5]. This physiological response manifests through the exercise-induced GIS commonly reported in athletes. Lower GIS include abdominal cramps, flatulence, urge to defecate, rectal bleeding, and diarrhea. Upper GIS include nausea, vomiting, regurgitation, chest pain, heartburn, and belching [6]. These exercise-induced GIS have the potential to impede athletic performance and can hinder an athlete’s ability to participate in or finish a race. 

Irritable bowel syndrome (IBS) is a chronic GI functional disorder characterized through recurrent abdominal pain, as well as a change in bowel movements predominating as either diarrhea, constipation or both [7]. In contrast, inflammatory bowel disease (IBD) is a relapsing inflammatory disease and includes both ulcerative colitis (UC) and Crohn’s disease (CD). UC is localized primarily in the large intestine, whereas CD can affect any area along the GI tract. GIS vary by individual and disease form, but often include abdominal pain, diarrhea, and rectal blood loss [8]. These symptoms can influence an athlete’s quality of life, health, and the ability to consistently train and compete.

Athletes with IBS or IBD make up a unique subset of the athletic population. Killian & Lee [9] found that an estimated 10% of endurance runners have IBS, if those who have been medically diagnosed and those diagnosed using the Rome III diagnostic criteria are combined. The prevalence increased to 23% if those meeting the less stringent Manning criteria were included. The presence of these diseases can have a significant impact on athletic performance. Exercise at low intensity appears to be beneficial in managing symptoms experienced by both IBS and IBD [10,11,12]. Conversely, as the intensity and duration of the exercise increases, exercise-induced GIS are often exacerbated, at least in healthy runners [2,3]. Unfortunately, evidence pertaining to the impact of high-intensity physical activity on an already compromised GI integrity and function in athletes with chronic GI disorders is scarce. 

Gastro-esophageal reflux disease (GERD) tends to be elevated in the Western world with a prevalence of 10–20% [13]. Potential mechanisms include increased gastric acid secretion, impaired gastric emptying, decreased esophageal clearance, a relaxed lower esophageal sphincter, and increased pressure gradient between the stomach and esophagus [14]. Furthermore, for runners in general, heartburn/reflux is a frequently reported upper GI symptom, with an incidence of ~10% in the population of healthy, Canadian runners [15]. Exercise may contribute to an individual experiencing heartburn with the severity being proportional to the type of exercise and its intensity [14]. Running appears to be particularly problematic with respect to gastro-esophageal reflux, or heartburn, as it was found to induce more exercise-related reflux compared to cycling [14].

Nutrition can have a strong influence on the management of exercise-induced GIS [3,16], as well as the management of symptoms related to chronic GI disorders [17,18,19,20,21]. Current sport nutrition guidelines provide limited guidance on food choices pre-exercise [22]; however, athletes report avoiding certain foods prior to exercise in hopes of preventing exercise-induced GIS symptoms. Two common dietary practices to prevent GIS are to follow a gluten-free diet or adhere to a low fermentable oligo-, di-, mono-saccharides and polyols (FODMAP) diet. Lis et al. [23] found that 41% of non-celiac athletes followed a gluten-free diet 50% of the time due to its belief of lowering GIS symptoms. In addition, athletes without GI diseases avoid at least one FODMAP food for improvements in exercise-induced GIS [16,24,25,26], with lactose being the most commonly avoided FODMAP for management of GIS symptoms [27]. In regards to IBD, preliminary research indicates some nutraceuticals and functional foods may minimize symptoms of IBD, but further research is needed in this area [21]. There is emerging evidence for a low FODMAP diet to improve GIS symptoms in non-athletic populations with IBS [17,18,20]. The role of diet in managing chronic reflux is inconclusive; however, caffeinated beverages are often cited as an aggravating factor [28]. Research examining pre-meal food choices in the subgroup of athletes with chronic GI disorders is limited. 

As nutrition plays a key role in managing exercise-induced GIS symptoms, examining the practices of athletes diagnosed with gastrointestinal diseases and their present symptoms will provide insight on this unique subset of athletes. The purpose of this study is to analyze voluntary, pre-exercise food restrictions and choices related to running induced GIS in athletes with IBS/IBD and heartburn/reflux. This study is a sub-set of a larger population of runners from a study examining dietary restrictions in runners via a validated and reliable questionnaire [29]. These findings will support the development of pre-exercise guidelines on specific food practices to manage exercise-induced GIS in athletes with GI diseases. 

## 2. Materials and Methods

### 2.1. Participants

The questionnaire was administered to endurance runners 18 years of age or older. Athletes were recruited from running groups, races, and at pre-race events across southern Alberta, Canada. Individuals with no GI-related medical conditions were analyzed separately [15]. Here we focus our analysis on participants with inflammatory bowel disease (IBD) or irritable bowel syndrome (IBS) (*n* = 53), and heartburn/reflux (*n* = 37). A control group of participants (*n* = 375) without any medical conditions or food allergies was included for comparison. The study received ethical approval from the Mount Royal University Human Research Ethics Board (ethics ID 2016-38). All participants provided written, informed consent. 

### 2.2. Questionnaire 

The researchers approached participants either at a running clinic, running event package pick-up or after completing a running event and asked them to complete a paper version of the questionnaire. The questionnaire was developed to collect information on basic demographics, running experience, medical conditions, voluntary food restrictions pre-racing, pre-race food choices, exercise-induced GIS experienced if they consume a trigger food, reasons for avoiding foods, and sources of information. The questionnaire asked participants to select options for each question by checking boxes. There was also the option for an open-ended “other”. The questionnaire was validated by experts in the field and tested for reliability in a test/re-test manner with 39 participants [29]. A copy of the questionnaire is available in Appendix A: Food Restriction in Running Questionnaire. 

### 2.3. Statistical Analysis 

Athlete responses were categorized into groups based on IBS/IBD, reflux, or no medical conditions. Performance level was categorized as lower recreational defined as “don’t compete” or “lower half of age group”; upper recreational defined as “upper half of age group”; and competitive athletes defined as “provincial, national or international”. Race distances were categorized as “don’t compete”; “5 km”; “6 to 10 km”; “11 to ½ marathon (21 km)”; “>½ marathon (>21 km)”. For determination of food avoidances pre-running, those who reported a specific food allergy were removed from the analysis of that particular food to ensure that the food avoidance was not related to allergies. Responses to the open-ended questions regarding pre-race food choices were coded into categories Appendix A: Customized Food Categories. Significant differences between the IBS/IBD or reflux groups and control group containing runners with no medical conditions or food allergies were determined by a Fisher’s exact test with *p* < 0.05 considered statistically significant. Frequencies for pre-race food choices and sources of information were determined. Data were analyzed using SPSS statistical software version 25 (IBM, Armonk, New York, NY, USA). Data set can be accessed at https://doi.org/10.5683/SP2/THOKY9.

## 3. Results

### 3.1. Participant Demographics

Five hundred and thirty runners participated in the study. Of those, 10% reported either IBD or IBS and 7% reflux. Five participants reported both reflux and IBS/IBD. Of those with IBS/IBD, 45 reported IBS, five reported IBD, and three reported both conditions. A physician was identified as the individual who provided the diagnosis of reflux or IBS/IBD in ~90% of the cases. The remaining 10% were either self-diagnosed, or diagnosed by a naturopath or allied health professional. To obtain a healthy group for comparison, 70 runners with other medical conditions and food allergies were removed from the data, and the analysis was based on the remaining 375 for the control group. The average age of those with IBS/IBD was 42(13) years, those with reflux 40(11) years and runners with no medical conditions 42(13) years. Participant characteristics are outlined in Table 1.

### 3.2. Foods Generally Not Well Tolerated

Foods perceived by runners as generally not well tolerated were determined via the question “Do you have any food intolerances (foods that result in negative symptoms other than an allergy) not related specifically to running?” and are presented in Table 2. In general, foods identified as problematic were significantly higher in the IBS/IBD group as compared to the general population. The reflux group reported increased frequencies of problems with grains, dairy, and coffee and tea as compared to the general population of runners. 

### 3.3. Food Avoidances Pre-Racing

Foods avoided in the pre-race period were assessed using the question “When RACING are there any types of food/fluid that you AVOID in your pre-race MEAL or SNACK (0–4 h before running RACES/COMPETITIONS?)”. There were no significant differences in the frequency of food avoidances between those with reflux and the control group. Legumes, soy milk, hot cereal, and grains containing gluten were avoided significantly more often in the IBS/IBD group as compared to the control group. The most commonly avoided foods are presented in Table 3. 

### 3.4. Foods Consumed Pre-Racing

Foods consumed in the pre-race period were evaluated by the question “When RACING please describe what you would typically choose to eat and/or drink (including any supplements or special products) within the 4h before running RACES/COMPETITIONS”. Grains containing gluten, high FODMAP fruits, and water were the most frequent choices for those with IBS/IBD. Foods reported to be consumed pre-race in 10% or more of runners with IBS/IBD are reported in Figure 1.

In those with reflux, grains containing gluten, eggs, and water were the most frequent choices for their pre-race meal. Foods consumed in the pre-race meal in ~10% or more of runners with reflux are reported in Figure 2. 

### 3.5. Exercise-Induced Gastrointestinal Symptoms

Symptoms runners experience during endurance running were determined by the question “When RACING what symptoms might you experience during a running RACE/COMPETITION if you had consumed a food that you typically avoid (identified above) before competing?”. In general, those with IBS/IBD reported the highest frequency of symptoms, followed by runners with reflux and the control runners having the fewest symptoms. Symptom prevalence is presented in Table 4.

### 3.6. Sources of Nutrition Information

Sources of nutrition information were assessed using the question “Where do you receive information about nutrition for running?”. The top five sources of information for those with IBS/IBD were internet (49%), magazines (40%), family/friends (38%), other athletes (32%), and Registered Dietitian (30%). In the group with reflux, the primary sources were internet (58%), magazines (39%), other athletes (39%), Registered Dietitian (33%), and teammates (31%). Furthermore, 36% of both groups reported that they had previously attended a nutrition workshop.

## 4. Discussion

The research presented here is significant, as to our knowledge, it is the first assessment of pre-exercise dietary restrictions and food choices in runners with IBS/IBD or reflux. The results provide critically needed information for athletes and health-care practitioners, and highlight management strategies athletes have developed in the absence of clear pre-exercise nutrition guidelines. It provides direction on potential dietary strategies for future investigation in randomized clinical trials. 

### 4.1. Characteristics of Runners with Medical Conditions

The prevalence of IBS/IBD in our runners was significantly greater in females, which aligns with other studies consistently reporting a greater prevalence of IBS in women compared to men, independent of the diagnostic criteria used [9,18,30,31,32]. While the prevalence of reflux was higher in female runners, it did not reach significance. When considering running distance and performance level, there were no differences between the groups with medical conditions as compared to the control group. It would be interesting for future research to consider if there are distances or intensities that are better suited for those with GI conditions.

Foods perceived as not well tolerated were significantly higher in the IBS/IBD group as compared to the runners in the control group. The data clearly demonstrates that those with IBS/IBD are more careful in their food choices and likely adhere to a more restrictive diet to manage their symptoms. Others support a high number of food intolerances in IBS/IBD [17,20]. Furthermore, medical professionals often advise patients with IBS/IBD to adhere to specific diets and/or have patients avoid offending foods to mitigate GIS [17,18,33]. In our study, grains containing gluten were avoided more often in the IBS/IBD cohort, which is in agreement with others who find a gluten-free diet improves symptoms in those with IBS [34]. Our results are also in parallel with others who report lactose containing foods and other high-FODMAP foods as common intolerances in those with IBS/IBD [18,34]. 

While the frequencies and number of food intolerances in those with reflux were not as high as in those with IBS/IBD, higher intolerances to grains, dairy, and coffee/tea were reported. Intolerances to coffee/tea would be expected, as these drinks are known to aggravate reflux and avoidance is often recommended [28]. Less evidence is available regarding the potential for gluten-containing grains to affect reflux; this is an area for future research. Conversely, this may simply reflect a general increase in gluten avoidance due to dietary attitudes and beliefs [35]. 

### 4.2. Food Avoidances in Runners with Medical Conditions 

Gastrointestinal symptomology can be affected by a plethora of nutritional factors; therefore, runners with IBS/IBD and reflux may adhere to stricter diets in an attempt to avoid exacerbated GIS. The most commonly avoided foods were milk products, high-protein, and high-fibre foods, all of which align with the recommendations to limit consumption of protein, fat, and fibre before exercise [22,36,37].

Over one half and one third of runners with IBS/IBD and reflux reported avoiding milk products prior to running, respectively. Dairy products contain a number of nutrients that potentially trigger GIS in this group of runners, including FODMAPs, protein, and fat [16,22]. Reduced intakes of dairy have been found in individuals with IBS, which is attributed to the belief that lactose intolerance exacerbates their symptoms [38], although good quality evidence supporting a milk-free diet to control symptoms of IBS is lacking [17]. Typical symptoms of lactose intolerance are further confounded by functional gastrointestinal disorders [39] and, in runners, the additional stress on the integrity and function of the GI tract due to exercise [40]. Not all dairy products contain lactose and when comparing milk products and lactose-free milk, there was a large reduction in avoidance for the latter. Future studies should investigate the subdivision of lactose-containing and lactose-free products to determine their effects on GIS [15], in addition to the impact of pre-run protein and fat specifically on GIS.

High-fibre foods are avoided by runners with IBS/IBD and reflux prior to running. This has also been observed in the general running population [15]. Prior to our research, it has been reported that endurance runners limit dietary fibre intake [41,42] albeit not specifically pre-exercise. Low-fibre intake prior to exercise is recommended [22,36] based on evidence of dietary fibre causing intestinal cramps [43]. Importantly, different types of fibre have a variety of actions on the gut including regulating gastrointestinal transit time and glucose homeostasis [44]. The role of the varying types of dietary fibre in the exacerbation and control of IBS is complex. However, evidence suggests no improvement in IBS symptoms with the inclusion of wheat bran, cereals, and fruit [17]. Food components that contain dietary fibre, including but not limited to cereals, fruit, and vegetables, have been associated with IBD symptoms. Conversely, consensus guidelines on the management of IBD in adults encourage a varied diet that includes dietary fibre [20]. High-fibre foods were also avoided by the reflux cohort pre-race; yet, should not be avoided completely as psyllium fibre has been demonstrated to decrease heartburn [45]. Optimally, an athlete should consider both the risk factors and benefits of specific high-fibre foods consumed prior to a run. The symptom profile should also be considered [17]. For runners with GI related medical disorders, the timing of fibre intakes will be critical to maximize health and performance. Future research regarding the impact of specific fibres on GIS in runners with IBS, IBD, and reflux would be helpful to establish evidence-based links between food constituents and GIS in this group. 

We have previously reported and discussed the voluntary avoidance of protein-rich foods pre-run in a healthy population of runners [15]. Our current study reveals that a significant number of runners with IBS/IBD and heartburn avoid meat, a high protein food, prior to runs, which is consistent with practice guidelines to limit protein intake immediately before exercise [22]. Furthermore, proteins from animal sources have been associated with increased reflux in non-athletes, which may partially explain their avoidance in runners [46]. Investigation into protein consumption prior to exercise remains scarce [47]; however, it has been associated with a greater risk of inducing GIS [3]. Indeed, in a study by Rehrer et al. [43], triathletes were more likely to develop and experience GIS after consuming protein during a half-Ironman triathlon. Rothschild et al. [47] report only 3.9% of endurance athletes consume high protein foods, such as eggs or protein powder prior to morning exercise. Additionally, 15 g of whey protein hydrolysate administered pre-exercise and every 20 min during running appears to increase GIS, suggesting difficulties in tolerating protein [48]. Cow’s milk protein may be associated with IBS symptoms. More specifically, A1 beta-casein has been linked to GIS in those with IBS, and this has not been observed with A2 beta-casein [17]. Dietary components that include protein have been associated with IBD, but those who suffer from IBD are advised to consume protein-rich foods to help contribute to nutrient requirements [20]. Expert consensus recommends 1.0 g protein/kg body weight/day for individuals with IBD in remission and 1.2 to 1.5 g protein/kg body weight/day for individuals with active UC or CD [49]. Therefore, timing of protein intake in runners with IBD is critical. 

One quarter of participants report they avoid coffee or tea prior to running. Caffeine is found in both coffee and tea, and as an ergogenic aid, caffeine is often praised for its performance-enhancing abilities. However, caffeine directly affects GI function by increasing gastric acid secretion and colonic motor activity, which can induce or worsen IBS symptoms [17]. This may be further impacted by running induced symptoms, as an increase in lower GIS has been associated with morning caffeine consumption in triathletes [50]. Additionally, caffeine is often touted as a trigger for heartburn/reflux [13]. Aside from its GI impact, caffeine can cause potential side-effects such as jitteriness, anxiety, and increased heart rate, which may be detrimental to performance [51]. Runners also want to avoid caffeine’s diuretic effects; however, the effects may be minimal when caffeine is habitually consumed in moderate (e.g. <180 mg) amounts [22]. Our research demonstrates that although one quarter of runners avoid coffee or tea prior to a run, many runners reported consuming coffee and/or tea before running. Therefore, dividing this category into “decaf coffee/tea” and “regular coffee/tea” may be of interest for future investigation. Other foods containing caffeine were also frequently avoided including chocolate and energy drinks; however, these are complex multi-ingredient foods, and the effects cannot be conclusively linked to their caffeine content. For example, while the most common ingredient in energy beverages is caffeine [52]; high-fructose corn syrup, which is a high FODMAP, is often used as a sweetener. This group of drinks are documented as a source of GIS in the healthy population [53]. Finally, chocolate and caffeine have previously been reported as trigger foods for heartburn/reflux [13]. All considered, it would be prudent for runners suffering from IBS/IBD and reflux to use these drinks with caution in an effort to avoid unnecessary GIS. 

Implementing a low FODMAP diet has been recognized as a treatment strategy for IBS and is an emerging strategy for those suffering from IBD [20]. It is also used by athletes to mitigate exercise-induced GIS [16]. We observed that high FODMAP foods such as milk products, soymilk, certain legumes, vegetables, high-fibre foods, and cold cereals (e.g. bran) were highly avoided pre-run. FODMAPs found within these foods can be poorly absorbed by the small intestine and may trigger or increase the magnitude of exercise associated GIS [16]. Consequently, it is possible that our research participants are reducing their FODMAP intake pre-run to avoid GIS potentially linked to both IBS/IBD related symptoms and the impact of running. While it has been well established that carbohydrate intake leading up to a race can benefit performance [22,54,55], runners suffering from IBS/IBD and reflux would benefit from specific carbohydrate-focused guidelines that can support performance whilst minimizing GIS. It is recommended that runners with IBS/IBD work with a Registered Sports Dietitian with expertise in the low FODMAP diet if they are struggling with exercise-induced GIS. The individual nature of each runner’s pre-race nutrition is critical, as mechanisms by which food related symptoms develop, and which dietary interventions are effective and safe remain unclear [20].

### 4.3. Foods Chosen Pre-Race in Runners with Gastrointestinal Diseases

Excluding water, only three of the top five foods reported to be consumed by runners with IBS/IBD or heartburn were carbohydrate-rich (i.e., grains with gluten, high FODMAP fruit, and hot cereal). Similarly, Rothschild et al. [47] found that approximately 36% of endurance male and female athletes indicated consuming carbohydrate-rich foods before almost every workout. The current sport nutrition recommendations for pre-exercise nutrition advocates for choices that are carbohydrate-rich, low-fat, low-fibre and low to moderate protein [22]. In our study, while many high-protein meats were avoided, we found slower digesting, high-protein and high-fat foods such as eggs, protein supplements, and nuts were reported as popular pre-race foods by runners with GI issues. This highlights the need for additional research regarding the amounts and sources of fat/protein that work best in the pre-race nutrition. Furthermore, it was surprising to see the runners with IBS/IBD to have chosen foods rich in FODMAPs as popular items, as high FODMAPs are generally associated with GIS in athletes [16,24,25,26]. Additionally, as previously mentioned, a low FODMAP diet is advocated to manage IBS/IBD in general [17,20]. As some FODMAP foods were consumed and others avoided, research should focus on the specific sources and quantity of FODMAPs in the pre-exercise meal. The reported sources for dietary advice may explain why runners with IBS/IBD chose pre-race foods that contradict nutrition guidelines to minimize GIS. The internet, magazines, family/friends and “other” athletes may not be reliable sources of nutrition advice, especially in regards to pre-exercise nutrition. With respect to reflux, these runners described that their pre-running food preferences were: water, grains, eggs, hot cereal, nuts and fruits rich in FODMAPs—foods currently not associated with reflux issues [56]. As with runners with IBS/IBD, those reported to suffer from reflux indicated that they were most likely to have their nutrition influenced by the internet, magazines and other athletes, as well as Registered Dietitians. 

### 4.4. Exercise-Induced Gastrointestinal Symptoms in Runners with Gastrointestinal Diseases 

Endurance athletes often experience a wide range of exercise-induced GIS [3], with running often being associated with the most prominent increases in GIS [2]. Here we find that runners with underlying gastrointestinal conditions were at a heightened risk of exercise-induced GIS as compared to control runners. Further investigation into the mechanisms of action in those with IBS/IBD and reflux would be valuable to better understand the relationship between a pre-existing condition and the additional stress of running on the GI tract. At least one third of individuals with inactive IBD suffer from coexisting bowel symptoms including bloating, flatulence, and diarrhea [20] and these symptoms can be mistaken for active IBD. Thus, there is the potential that these GIS can be erroneously attributed to exercise. In the future, it could be valuable to closely examine the timing, frequency, and severity of GIS in this demographic to clarify the link between exercise and nutrition.

The most common exercise-induced GIS experienced by runners diagnosed with IBS/IBD, if they were to consume an offending food, were ‘stomach pain/cramps,’ ‘intestinal pain/discomfort,’ ‘diarrhea,’ and ‘bloating.’ When compared to the general running population, these individuals were significantly more likely to experience these GIS while running. Echoing IBS/IBD, runners diagnosed with reflux most commonly experienced ‘stomach pain/cramps,’ followed by ‘bloating,’ ‘reflux/heartburn,’ and ‘diarrhea.’ These findings align with our previously published study assessing GIS experienced in the general running population, wherein ‘stomach pain/cramps’ was the most commonly reported GIS [15]. Similar results were also found by ten Haaf et al. [57], with ‘side ache’ being the number one experienced GIS in female runners. Notably, however, ‘side ache’ is regional and may not be purely gastrointestinal in cause. In the current study, ‘gas’ and ‘urge to defecate’ were also commonly reported, consistent with other studies assessing GIS in runners [2,57,58,59]. 

GIS are problematic within themselves, but can also serve as an underlying risk of underperformance and event dropout. Implications on performance depend not only on nutritional factors, but exercise duration and intensity. ter Steege et al. [60] found ‘severe side stitch’ as a common GIS, although prevalence was based on distance ran, as it was significantly more prevalent in the 10 km group compared to the 42 km group. Interestingly, in the same study, women were significantly more likely to experience ‘severe side stitch’ than men were. While the current study did not report on GIS severity, previous studies have reported rating GIS severity using Likert or 10-point scales. Studies rating GIS using both a 0–10 Likert scale [2] and a 0–9 scale [61] have reported symptoms scoring ≥5 are likely to be ergolytic for performance. 

### 4.5. Limitations 

The data presented here is based on a self-report questionnaire. While it cannot offer evidence-informed guidelines, it does provide researchers with specific foods that can be tested for their ability to cause or prevent exercise-induced GIS. Another limitation of this research is that it does not link a specific food to a specific symptom, so only generalizations can be made regarding pre-exercise offending foods and GIS during exercise. With respect to IBS/IBD, we combined these groups due to the small number of runners with IBD; however, acknowledge that these are two different disorders and ideally should be considered separately in future research. While there are similarities in triggering foods between IBS and IBD, there may also be differences. Furthermore, the questionnaire did not assess disease activity, and it should be noted that the responses may vary depending on if the individual is in remission or actively experiencing symptoms. Finally, while those with food allergies were removed from the analysis of a particular food, those with foods not well tolerated were included. This can make it difficult to tease out if the food was avoided only pre-running. We did not ask participants if they were following a low FODMAP diet or any other dietary pattern outside of the pre-race meal. This would be of interest in future studies. It was also apparent that there was confusion between food allergies and intolerances, further complicating the assessment. A comparison of the rates of foods not generally well tolerated (Table 2) to those avoided pre-race (Table 3); however, can provide further insight to the avoidance in specifically the pre-running nutrition.

## 5. Conclusions

Runners with IBS/IBD and reflux disease experience greater exercise-induced GIS than the control runners with no medical conditions or allergies. One potential strategy to help mitigate GIS in runners with medical conditions is to alter the pre-exercise food choices. Our results suggest future research should focus on testing dairy, caffeinated products, gluten containing grains, and high FODMAP foods, in the pre-exercise nutrition, as potentially offending foods. While high-fibre and high-protein foods are flagged in the current recommendations, additional research regarding the timing and sources of these foods would be beneficial, as they are important in the diet in general. 

## Figures and Tables

**Figure 1 nutrients-13-00448-f001:**
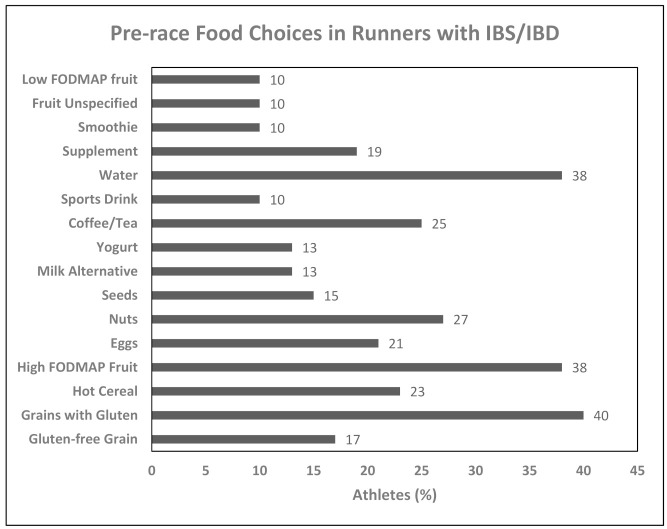
Commonly consumed foods pre-race for runners with irritable bowel syndrome (IBS) or inflam-matory bowel disease (IBD).

**Figure 2 nutrients-13-00448-f002:**
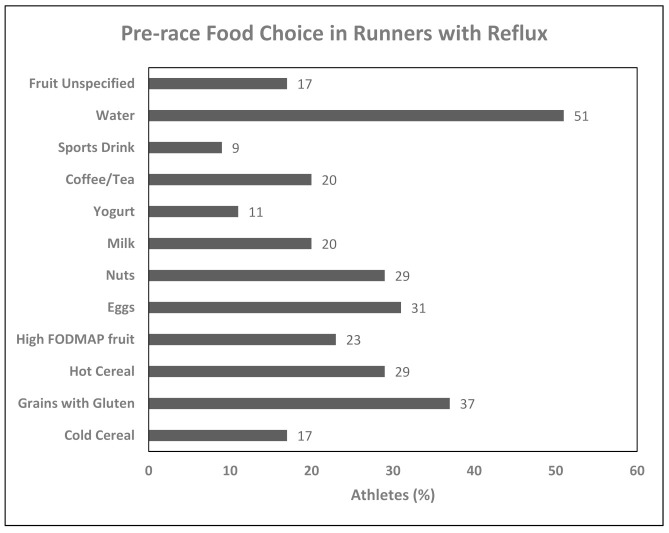
Commonly consumed foods pre-race for runners with reflux.

**Table 1 nutrients-13-00448-t001:** Participant characteristics.

Participants	IBS/IBD *n* (%)	Reflux *n* (%)	Control Runners *n* (%)
Gender			
Male	2 (4) (<0.001)	11 (31)	162 (45)
Female	50 (96)	25 (69)	195 (55)
Performance Level			
Lower Recreational	25 (49)	18 (50)	128 (35)
Upper Recreational	24 (47)	16 (44)	208 (56)
Competitive	2 (4)	2 (6)	33 (9)
Run Distance			
Don’t Compete	1 (2)	0 (0)	0 (0)
5 km	13 (25)	9 (26)	63 (18)
6–10 km	16 (31)	15 (43)	123 (35)
Half-marathon	14 (27)	8 (23)	130 (37)
Marathon	8 (15)	3 (9)	40 (11)
Allergies			
Tree Nuts	4 (8)	2 (5)	n/a
Milk	12 (23)	4 (11)	n/a
Soy	2 (4)	2 (5)	n/a
Sulfites	3 (6)	1 (3)	n/a
Whey	5 (9)	1 (3)	n/a
Eggs	2 (4)	0 (0)	n/a
Peanuts	3 (6)	1 (3)	n/a
Fish	1 (2)	5 (14)	n/a
Wheat	6 (11)	0 (0)	n/a
Gluten	8 (15)	1 (3)	n/a
Casein	5 (9)	1 (3)	n/a
Avocados	1 (2)	0 (0)	n/a
Corn & oats	1 (2)	0 (0)	n/a
Cranberries & pineapple	1 (2)	0 (0)	n/a
Garlic	1 (2)	0 (0)	n/a
Octopus	1 (2)	0 (0)	n/a
MSG	0 (0)	1 (3)	n/a
Oranges	0 (0)	1 (3)	n/a
Horseradish	0 (0)	1 (3)	n/a
Bananas & capers	0 (0)	1 (3)	n/a

Participant characteristics of those with IBS/IBD, reflux/heartburn and control runners with no medical conditions or allergies. Inflammatory bowel disease (IBD), irritable bowel syndrome (IBS), monosodium glutamate (MSG) Five participants reported both heartburn and IBS/IBD and their results were included in the analyses for both categories. Differences between those with IBS/IBD or reflux and the control group with no medical conditions or allergies were determined by a Fisher’s Exact test. *p*-values are provided next to the percentages; *p* < 0.05 was considered significant.

**Table 2 nutrients-13-00448-t002:** Foods Not Well Tolerated.

Foods Not Well Tolerated	IBS/IBD *n* (%)	Reflux *n* (%)	Control Runners *n* (%)
Grains	18 (34) (<0.001)	6 (16) (0.014)	18 (5)
Legumes	8 (15) (<0.001)	0 (0)	4 (1)
Soy milk	7 (13) (<0.001)	2 (5)	3 (1)
Starchy vegetable	5 (9) (<0.001)	0 (0)	2 (0.5)
Cold cereal	5 (9) (0.010)	2 (5)	7 (2)
Hot cereal	7 (13) (<0.001)	1 (3)	3 (1)
Nuts	4 (8) (0.006)	2 (5)	3 (1)
Coffee/tea	6 (11) (0.001)	3 (8) (0.039)	6 (2)
Yogurt	17(32) (<0.001)	8 (22) (0.001)	17 (5)
Eggs	9 (17) (<0.001)	1 (3)	4 (1)
Cheese	17 (32) (<0.001)	11 (30) (<0.001)	23 (6)
Milk	26 (49) (<0.001)	15 (41) (<0.001)	52 (14)
Sports Bar/gel	5 (9) (0.037)	2 (5)	11 (3)
Vegetable	8 (15) (0.001)	3 (8)	12 (3)
Gluten-free grains	7 (13) (0.001)	1 (3)	9 (2)
Energy drinks	4 (8) (0.036)	2 (5)	7 (2)
Sports drink	3 (6)	1 (3)	5 (1)
Fruit	4 (8) (0.025)	2 (5)	6 (2)

Commonly self-reported foods not well tolerated in runners with IBS/IBD, reflux/heartburn, and the control runners with no medical conditions or allergies. Inflammatory bowel disease (IBD), irritable bowel syndrome (IBS). Five participants reported both reflux and IBS/IBD and their results were included in both categories. Differences between those with IBS/IBD or reflux and the control group with no medical conditions or allergies were determined by a Fisher’s Exact test. *p*-values are provided next to the percentages; *p* < 0.05 was considered significant.

**Table 3 nutrients-13-00448-t003:** Foods Avoided Pre-race.

Foods Avoided	IBS/IBD *n* (%)	Reflux *n* (%)	Control Runners *n* (%)
Meat	16 (31)	13 (36)	121 (33)
Milk products	21 (53)	12 (38)	135 (37)
Fish/seafood	11 (22)	6 (19)	109 (30)
Poultry	11 (21)	9 (25)	93 (25)
High-fibre	14 (27)	12 (33)	85 (23)
Chocolate	12 (23)	9 (25)	83 (23)
Legumes	19 (37) (0.006)	11 (31)	70 (19)
Coffee/tea	13 (25)	9 (25)	71 (19)
Energy drinks	11 (21)	3 (8)	61 (17)
Starchy vegetable	8 (15)	4 (11)	56 (15)
Lactose-free milk	4 (10)	2 (7)	55 (15)
Eggs	9 (18)	5 (14)	54 (15)
Soy milk	14 (28) (0.012)	6 (18)	50 (14)
Sports drink	5 (10)	1 (3)	42 (11)
Vegetables	7 (14)	7 (19)	44 (12)
Cold cereal	7 (16)	6 (17)	44 (12)
Grain	14 (32) (<0.001)	6 (17)	37 (10)
Hot cereal	9 (21) (0.010)	3 (9)	28 (8)
Gluten-free grain	6 (12)	4 (11)	28 (8)
Sports bar/gel	6 (12)	3 (8)	39 (11)
Juice	8 (15)	3 (8)	37 (10)
Smoothie	6 (12)	6 (17)	33 (9)
Fruit	6 (12)	5 (14)	30 (8)
Nuts	6 (13)	5 (15)	40 (11)
Coconut milk	7 (14)	4 (11)	44 (12)
Almond milk	4 (9)	3 (9)	41 (11)

Foods commonly avoided pre-racing in runners with IBS/IBD, reflux/heartburn, and control run-ners with no medical conditions or allergies. Inflammatory bowel disease (IBD), irritable bowel syndrome (IBS). Five participants reported both reflux and IBS/IBD and their results were included in both categories. Differences between those with IBS/IBD or reflux and the control population of runners was determined by a Fisher’s Exact test. *p*-values are provided next to the percentages; *p* < 0.05 was considered significant.

**Table 4 nutrients-13-00448-t004:** Symptoms Experienced if Consumed an Offending Food Pre-race.

Symptoms	IBS/IBD *n* (%)	Reflux *n* (%)	Control Runners *n* (%)
Stomach pain/cramps	40 (77) (<0.001)	19 (53)	157 (43)
Intestinal pain/discomfort	30 (58) (<0.001)	12 (33)	85 (23)
Side ache/stitch	12 (23)	11 (31)	81 (22)
Urge to defecate	16 (31)	13 (36)	79 (21)
Bloating	27 (52) (<0.001)	18 (50) (<0.001)	72 (20)
Diarrhea	30 (58) (<0.001)	14 (39) (0.007)	66 (18)
Fullness/heaviness	11 (24)	7 (25)	60 (18)
Burping/belching	10 (19)	7 (19)	58 (16)
Gas	18 (35) (0.003]	11 (31) (0.039)	60 (16)
Nausea/vomiting	15 (29) (0.001]	11 (31) (0.002)	38 (10)
Reflux/heartburn	6 (12)	15 (42) (<0.001)	26 (7)
Bleeding	0 (0)	0 (0)	1 (0.3)

Exercise-induced gastrointestinal symptoms experienced by runners while racing if they consume an offending food. Inflammatory bowel disease (IBD), irritable bowel syndrome (IBS). Percentages represent the frequency of individuals from each category who report they would experience the symptom if they ate a food they would typically avoid. Five participants reported both reflux and IBS/IBD and their results were included in both categories. Differences between those with IBS/IBD or reflux and the control population of runners was determined by a Fisher’s Exact test. *p*-values are provided next to the percentages; *p* < 0.05 was considered significant.

## Data Availability

The data presented in this study are openly available in Scholars Portal Dataverse at https://doi.org/10.5683/SP2/THOKY9.

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
