# Peer review of "Dietary Patterns in Runners with Gastrointestinal Disorders"

_nutrients, 2021, doi:10.3390/nu13020448_

Round 1

Reviewer 1 Report

This paper is interesting and well-written. However, I think that it should be improved. 1) Food intolerances self-reported have no relevance in the context of a scientific paper... Although the tests able to diagnose food intolerances are not standardized, at least a laboratory assessment should be performed to report a result. Please remove it from the results section and correct discussion accordingly 2) It is likely that a patient with IBD able to perfom physical activity is in remission, but disease activity of patients with IBD should be reported. If these data are not available, this should be reported in the discussion as a limitation. 3) With regard for patients with gastroesophageal reflux disease, there is evidence that vegetal proteins are associated with a lower number of refluxes, particularly acid refluxes, and with a reduced number of symptoms during the first postprandial hour (Martinucci et al. Gastroenterol Res Pract, 2018). This study should be discussed in discussion section, accordingly to the results of the present study 4) The impact of Low-FODMAP diet in the long-term is debated (Bellini et al, Nutrients, 2020). For this reason, I think that the duration of the low-FODMAP regimen should be reported in patients following a stable low-FODMAP diet. If these data are not available, this should be reported in the discussion as a limitation.

Author Response

Response to the Reviewers

We would like to thank the reviewers for their considered feedback. We have made the requested changes. Responses to each comment are provided below.

Reviewer #1

This paper is interesting and well-written. However, I think that it should be improved.

Thank you for your interest in our paper and suggestions. The manuscript is improved by your feedback.

1) Food intolerances self-reported have no relevance in the context of a scientific paper... Although the tests able to diagnose food intolerances are not standardized, at least a laboratory assessment should be performed to report a result. Please remove it from the results section and correct discussion accordingly

Thank you for the comment. We do recognize that food intolerances are much less clear than food allergies and there is a lack of standardized tests. We do feel, however, that there is value in reporting foods that are typically problematic for individuals with the gastrointestinal disorders mentioned. We also feel that it is important in the overall context of the manuscript to help the reader distinguished between the frequency of these food avoidances in general and specifically in the pre-run period. After careful consideration, we have opted to keep Table 2, however, have renamed it “Foods Not Well Tolerated” in light of the reviewer’s comments lines 165-172; Line 247.

2) It is likely that a patient with IBD able to perfom physical activity is in remission, but disease activity of patients with IBD should be reported. If these data are not available, this should be reported in the discussion as a limitation.

Yes, we agree that disease activity could influence the responses. Unfortunately, we did not explicitly assess disease activity. Given that the participants were completing the questionnaire after an event, the assumption would be they are in remission, however, we have noted this in the limitations lines 435-438.

3) With regard for patients with gastroesophageal reflux disease, there is evidence that vegetal proteins are associated with a lower number of refluxes, particularly acid refluxes, and with a reduced number of symptoms during the first postprandial hour (Martinucci et al. Gastroenterol Res Pract, 2018). This study should be discussed in discussion section, accordingly to the results of the present study

Thank you for the valuable reference. We have included a comment regarding vegetal vs animal proteins in lines 308-310 and included the reference in the reference list (#46).

4) The impact of Low-FODMAP diet in the long-term is debated (Bellini et al, Nutrients, 2020). For this reason, I think that the duration of the low-FODMAP regimen should be reported in patients following a stable low-FODMAP diet. If these data are not available, this should be reported in the discussion as a limitation.

Thank you for the comment. We did not assess adherence to a low-FODMAP diet, we noted that some of the foods avoided were high in FODMAP and this may partially explain why they were avoiding these types of foods. We did not specifically ask participants if they were following a low-FODMAP diet or other prescribed diet. This would be of interest for future studies. We have made a note of this in the limitations lines 440-442.

Reviewer 2 Report

Dear Sir I have read with great interest the manuscript entitled

Dietary patterns in runners with gastrointestinal disorders. The authors showed that runners affected by different gastrointestinal disorders consumes during the pre-race phase potentially dangerous food  

The manuscript is well written

Following my concerns

Line 55-57: Old criteria. Are there some studies with Rome IV ?

Line 144: In my opinion IBD and IBS should not be considered togheter

Line 146: And the other 10%?

Line 154-157: Delete, already explained

Line 159: It is not clear if these intolerances were diagnosed with test or just felt by the athlete

Line 168-171: See above

Line 183-185: See above

Line 191: 10%?

Line: 398: “side ache” it is not purely gastrointestinal, it could be a diaphragmatic issue, because it passes with the pressure of the point. It could skew the data.

Author Response

Response to the Reviewers

We would like to thank the reviewers for their considered feedback. We have made the requested changes. Responses to each comment are provided below.

Reviewer #2

I have read with great interest the manuscript entitled

Dietary patterns in runners with gastrointestinal disorders. The authors showed that runners affected by different gastrointestinal disorders consumes during the pre-race phase potentially dangerous food  

The manuscript is well written

Thank you for your interests and comments.

Following my concerns

 Line 55-57: Old criteria. Are there some studies with Rome IV ?

Unfortunately, there are very few studies regarding IBS in endurance runners specifically. While it would be nice to have papers using the more recent criteria, we were unable to find any.

Line 144: In my opinion IBD and IBS should not be considered together

We agree with the reviewer that these are two different disorders and it would be ideal to consider them separately, however, our sample did not allow for such separation. We only had 8 participants who reported IBD. Furthermore, there appears to be confusion among the participants regarding these disorders as 3 reported both conditions. We have included a comment on this in the limitations in lines 432-435.

Line 146: And the other 10%?

We have included information on the other means of diagnosis in lines 151-152.

Line 154-157: Delete, already explained

We agree that we explained this in the statistics section in the methods, however, were under the understanding that the tables needed to be able to stand-alone. We can certainly remove this portion of the table descriptions if the reviewer insists, however.

Line 159: It is not clear if these intolerances were diagnosed with test or just felt by the athlete

Thank you for your comment. Food intolerances are indeed a grey area, hence our distinction between intolerances and allergies in the questionnaire. We have provided clarity in lines 165 that these were not required to be diagnosed by standardized tests.

Line 168-171: See above

We agree that we explained this in the statistics section in the methods, however, were under the understanding that the tables needed to be able to stand-alone. We can certainly remove this portion of the table descriptions if the reviewer insists, however.

Line 183-185: See above

We agree that we explained this in the statistics section in the methods, however, were under the understanding that the tables needed to be able to stand-alone. We can certainly remove this portion of the table descriptions if the reviewer insists, however.

Line 191: 10%?

We provided clarity that the figure pertains to those foods avoided by 10% or more of the runners in line 204.

Line: 398: “side ache” it is not purely gastrointestinal, it could be a diaphragmatic issue, because it passes with the pressure of the point. It could skew the data.

Thank you we have included a comment regarding the regional nature of ‘side ache” and a note that it may not be purely gastrointestinal in lines 411-412.

Round 2

Reviewer 1 Report

The authors' reply is satisfying.